# Screening, Docking, and Molecular Dynamics Study of Natural Compounds as an Anti-HER2 for the Management of Breast Cancer

**DOI:** 10.3390/life12111729

**Published:** 2022-10-28

**Authors:** Sayed Sartaj Sohrab, Mohammad Amjad Kamal

**Affiliations:** 1Special Infectious Agents Unit, King Fahd Medical Research Center, King Abdulaziz University, Jeddah 22254, Saudi Arabia; 2Department of Medical Laboratory Sciences, Faculty of Applied Medical Sciences, King Abdulaziz University, Jeddah 22254, Saudi Arabia; 3Institutes for Systems Genetics, Frontiers Science Center for Disease-Related Molecular Network, West China Hospital, Sichuan University, Chengdu 610041, China; 4King Fahd Medical Research Center, King Abdulaziz University, Jeddah 22254, Saudi Arabia; 5Department of Pharmacy, Faculty of Allied Health Sciences, Daffodil International University, Dhaka 1207, Bangladesh; 6Enzymoics, 7 Peterlee Place, Novel Global Community Educational Foundation, Hebersham, NSW 2770, Australia

**Keywords:** natural compounds screening, docking, MD simulation, interaction study

## Abstract

Breast cancer (BC) is one of the most frequent types of cancer that affect women. Human epidermal growth factor receptor-2 (HER2) is responsible for 20% of all BC cases. The use of anti-HER2 natural compounds in the cure of BC that is HER2-positive patients has resulted in significant increases in survival in both early and advanced stages. The findings of in-silico research support the use of ligands as possible HER2 inhibitors, and molecules with high free energy of binding may have considerable anti-BC action, making them candidates for future drug development. The inhibitory activity of selected ligands like ZINC43069427 and ZINC95918662 against HER2 was found to be −11.0 and −8.50 kcal/mol, respectively. The amino acid residues Leu726, Val734, Ala751, Lys753, Thr798, Gly804, Arg849, Leu852, Thr862, and Asp863 were found in common interaction as compared to the control compound Lapatinib. Molecular dynamics study calculations of these selected potent inhibitors were conducted and found to be stable over the 50 ns simulation time in terms of root mean square deviation (RMSD), root-mean square fluctuation (RMSF), radius of gyration (Rg), and solvent accessible surface area (SASA). In addition, there are several parameters such as absorption, distribution, metabolism, and excretion toxicity (ADMET), physicochemical, and drug-likeness that were checked and found in good range to be potential lead-like molecules. Several drug-likeness rules like Lipinski, Ghose, Veber, Egan, and Muegge were checked and found to be positive for these rules. Based on these calculations and different parameters, these top two selected natural compounds can be used as potential candidates for anti-HER2 for the management of BC.

## 1. Introduction

Breast cancer (BC) is the most frequently diagnosed malignancy in women, and it is also the second leading cause of cancer-related death in women. The ability to detect and diagnose BC has significantly increased [1]. BC is commonly caused by a high-fat diet, excessive alcohol intake, and a lack of physical activity. The elimination of these factors may help to reduce morbidity and mortality. Breast self-examination, mammography, ultrasonography, and magnetic resonance imaging may help in the early diagnosis of tumors [2]. Female BC (11.7%) has surpassed lung cancer as the most often diagnosed malignancy, followed by lung (11.4%), colorectal (10.0%), prostate (7.3%), and stomach (5.6%) [3]. Every year, BC affects 2.1 million women, and it is the leading cause of cancer-related deaths in women. BC affected 2.3 million women globally in 2020, with 685,000 fatalities, accounting for ~15% of all female cancer deaths [4].

Human epidermal growth factor receptor 2 (HER2) overexpression has been linked to adenocarcinomas such as breast, ovarian, endometrial, cervix, and lung cancer [5]. As a result, HER2 is a significant target for several forms of cancer treatment [6]. Overexpression of the HER2 gene is seen in 15–20% of BC, and it is usually linked to a high level of biological and clinical disease aggressiveness [7,8]. HER2 plays an important role in the biology of many cancers [9]. Furthermore, innovative HER2-targeted medicines have been widely studied in recent years and have shown more positive outcomes [10].

Natural products are used to reduce the progression and symptoms of a wide range of ailments. These products have long been utilized as pharmaceuticals to treat a variety of human ailments. Natural lead compounds are being used as fresh templates for producing more effective and safer medicines [11]. Virtual screening (VS) is one of the most widely used methods for finding new scaffolds and lead compounds [12]. Molecular docking is a prominent method for VS that has been proven to be effective in identifying hits and optimizing leads [13,14]. Large chemical libraries have been effectively screened against therapeutic targets and identified using molecular docking. In-house and commercial libraries have both been screened using docking approaches. The docking-based screening was used in a recent investigation to find potential inhibitors of HER2 [15,16,17,18]. There have been several other studies where docking-based VS has been successful in identifying new inhibitors [19,20,21]. Overall, this study aimed to explore a library of natural compounds obtained from the ZINC database (https://zinc.docking.org, accessed on 23 September 2022) against HER2 for the management of BC.

## 2. Materials and Methods

### 2.1. Preparation of Protein

The crystal structure of the kinase domain of human HER2 was obtained from Protein Data Bank (PDB) with PDB ID: 3PP0 [22]. The structure was downloaded in .pdb format and was further prepared for the docking process. The missing side residues were filled using Swiss-PDB Viewer [23]. The final structure obtained was saved in .pdb format for further studies.

### 2.2. Screening for Compounds

Initially, 80617 natural compounds were obtained from the ZINC database and structures were downloaded [24] in SDF format for further screening procedures. Several filters were applied to the structures to select potential drug-like compounds. Initially, the compounds were filtered by Lipinski’s rule of five (RO5) [25]. The SWISS-ADME server was used to screen drug-like compounds via RO5 based on pharmacokinetics, drug-likeliness, and medicinal chemistry of small molecules [26]. In addition, the physicochemical properties and the drug ability of the selected compounds were subjected to Ghose filter [27], Veber [28], Egan [29], and Muegge Filter [30].

### 2.3. Validation of Docking Protocol

In the validation of docking protocols, the ligand such as positive control (Lapatinib, Afatinib, Sapitinib, and Salvianolic acid C) was docked against the active site of 3PP0. The amino acid residues, namely, Leu726, Val734, Ala751, Lys753, Thr798, Gly804, Arg849, Leu852, Thr862, and Asp863 were found in common interaction for these selected control against the target. By considering these amino acid residues as the active pocket, screening was performed and finally, ZINC43069427 and ZINC95918662 were selected as lead-like molecules. The visual examination of docking simulation outcomes shows that the employed scoring function was appropriate. Therefore, the result supported the hypothesis, and the binding energy was found to be almost in the same range.

### 2.4. Multiple Ligand Docking

For docking simulations, Autodock was used [31]. Water molecules and heteroatoms were initially eliminated from the structures [32]. The docking technique was completed after the inclusion of gasteiger charges and H-bond. Concerning the found active site, a grid box of 25 × 22 × 19 Å was allocated to the receptor’s surface. Multiple ligands have been tested against the receptor in this investigation. PyRx was used in the Autodock environment to perform multiple ligand docking [33]. The root mean square deviation (RMSD), lowest energy conformer, and hydrogen bond (HB) interaction of the docked structures were investigated.

### 2.5. Protein Plus Server

Protein Plus Server was used to create a 2D protein-ligand interaction profile [34].

### 2.6. MD Simulation

MD simulations were carried out for the HER2-ZINC43069427 and HER2-ZINC95918662 complex for a period of 50 ns using GROMACS v5.1 [35]. The Simple Point Charge (SPC) water model was used to solvate the unit cell, which was defined as a cubical box with a minimum distance of 10 Å from the protein surface. GROMOS96 53a6 force field was used to build the topologies of the target. The system was subjected to energy minimization before the MD run by using the steepest descent integrator for 5000 steps with a force convergence of 1000 kcal/mol/nm. Following that, NVT and NPT ensembles were used to equilibrate each protein-ligand combination. During equilibration, each system was connected to temperature and pressure controllers by Berendsen and Parrinello-Rahman, respectively, to maintain a temperature of 300 K and a pressure of 1 bar. The linear constraint solver (LINCS) approach was used to limit all bond lengths. [36]. The couplings with the thermostats were then relaxed for a 50 ns production run under the micro-canonical ensemble. During the production run, the coordinates were stored every 10 ps with a time step of 2 fs. The Grace 5.1.23 software created 2D graphs displaying the intrinsic dynamical stabilities of the complexes as measured by the RMSD, root-mean-square fluctuation (RMSF), the radius of gyration (Rg), and solvent accessible surface area (SASA).

## 3. Results and Discussion

Initially, 80,617 natural compounds were obtained from the ZINC database. Further, these compounds were screened using the RO5 and SWISS- absorption, distribution, metabolism, and excretion (ADME) before binding. Among them, the top 20 screened compounds were listed in Table 1, in which only the top two compounds such as ZINC43069427 and ZINC95918662 displayed the highest negative binding energy −11.0 kcal/mol and −8.50 kcal/mol, respectively, as shown in Table 1. The virtual screening process of a library having natural compounds obtained from the ZINC database was performed against HER2, as shown in Figure 1.

Swiss ADME was employed for the identification of ADME properties of the selected ligands such as ZINC43069427 and ZINC95918662. The criteria for drug-like molecules should be followed, such as absorption, distribution, metabolism, excretion, and toxicity constraints [37]. If the conditions are met, in silico computer investigations of these factors at an early stage of drug design provide timely and useful information for further exploration [38]. Human intestinal absorption (HIA), Caco-2 cell permeability, blood-brain barrier (BBB) penetration, and AMES test were identified for the selected ligands. The principal location for absorption of an oral medication is the intestine. The value of HIA for the selected ligands ZINC43069427 and ZINC95918662 were found to be 92.48 and 82.45%, respectively. There are several parameters such as ADMET, physicochemical, and drug-likeness, reported in Table 2.

Several drug-likeness rules, such as Lipinski [25], Ghose [27], Veber [28], Egan [29], and Muegge [39] rules, were checked and found to be positive, as shown in Table 3.

The standard ligand, Lapatinib, resulted in a docking score of −7.65 kcal/mol and was used to compare other natural compounds docked with the HER2 receptor. Lapatinib is a well-known drug for HER2 [10]. Afatinib, Sapitinib, and Salvianolic acid C were also used as positive controls along with Lapatinib for HER2 [40,41,42]. The free energy of binding for these controls were found to be −7.80, −7.15, and −7.10 kcal/mol, respectively. Among these four selected positive controls, only Lapatinib was used for in-depth study because it showed the highest negative value of free energy of binding. The two selected compounds, ZINC43069427 and ZINC95918662, were found to be better than the selected positive controls. The amino acid residues, Leu726, Val734, Ala751, Lys753, Thr798, Gly804, Arg849, Leu852, Thr862, and Asp863, were found in common interaction for these selected controls. The error bars of the binding free energies were computed from the standard deviation of multiple docking runs for the ligands ZINC43069427 and ZINC95918662, and Lapatinib. They were found to be −11.0 ± 0.14 and −8.5 ± 0.70, and −7.65 ± 1.41 kcal/mol for ZINC43069427 and ZINC95918662, and Lapatinib, respectively.

### 3.1. The Interaction of ZINC43069427 with HER2

The free energy of binding for the complex HER2-ZINC43069427 was found to be −11.0 kcal/mol. ZINC43069427 was found to interact with the HER2 receptor through Leu726, Val734, Ala751, Lys753, Ser783, Leu796, Thr798, Gly804, Cys805, Asp808, Arg849, Leu852, Thr862, and Asp863 amino acid residues. Among these amino acid residues, Asp808 was found to be involved in H-bond formation with the ligand ZINC43069427. Thr798, Val734, Thr862, and Leu852 were found to be involved in hydrophobic interactions. Cys805 showed electrostatic interactions. The Van der Waals interaction was also observed through the amino acid residues Ser783, Asp863, and Gly804 with ZINC43069427. Pi-alkyl and Pi-sigma were also found in the complex formation of HER2-ZINC43069427. Ala751, Leu726, and Leu796 were involved in the Pi-alkyl, while Val734 was found to be involved in Pi-sigma. The complex structure HER2-ZINC43069427 is shown in Figure 2A.

### 3.2. The Interaction of ZINC95918662 with HER2

The free energy of binding for the complex HER2-ZINC95918662 was found to be −8.50 kcal/mol. ZINC95918662 was found to interact with the HER2 receptor through Lys724, Leu726, Ser728, Gly729, Val734, Ala751, Ile752, Lys753, Thr798, Leu800, Met801, Pro802, and Gly804 amino acid residues. Among these amino acid residues, Asp836, Ser728, and Pro802 were found to be involved in H-bond formation with the ligand ZINC95918662. Leu852, Val734, Leu726, and Gly804 were found to be involved in the hydrophobic interaction. Pro802 and Ser728 showed electrostatic interaction. The Van der Waals interaction was also observed through the amino acid residues Leu800, Met801, Lys724, Cys805, Asp808, Gly729, Thr798, and Ile752 with ZINC43069427. Pi-alkyl and Pi-sigma were also found in the complex formation of HER2-ZINC43069427. Ala751 and Lys753 were involved in the Pi-alkyl, while Leu726 was found to be involved in Pi-sigma. The complex structure HER2-ZINC95918662 is shown in Figure 2B.

### 3.3. The Interaction of Lapatinib with HER2

The free energy of binding for the complex HER2-Lapatinib was found to be −7.65 kcal/mol. Lapatinib was found to interact with HER2 receptors through Lys724, Leu726, Val734, Lys736, Ala751, Lys753, Ser783, Leu785, Leu796, Thr798, Met801, Pro802, Gly804, Cys805, Asp808, Leu852, Thr862, and Asp863 amino acid residues. Among these amino acid residues, Lys736 and Pro802 were found to be involved in H-bond formation with Lapatinib. Thr798, Leu852, Val734, and Gly804 were found to be involved in the hydrophobic interaction. Asp863 and Lys724 showed electrostatic interaction. The Van der Waals interaction was also observed through the amino acid residues Phe864, Thr798, Ser783, and Met801 with Lapatinib. Pi-alkyl and Pi-sigma were also found in the complex formation of HER2-Lapatinib. Cys805, Lys753, and Leu796 were involved in the Pi-alkyl, while Leu726 was found to be involved in Pi-sigma. The complex HER2-Lapatinib is shown in Figure 2C.

The selected ligands ZINC43069427 and ZINC95918662 were showing more negative free energy of binding as compared to the control drug Lapatinib. The free energy of binding for the selected complexes HER2-ZINC43069427, HER2-ZINC95918662, and control HER2-Lapatinib were found to be −11.0, −8.5, and −7.65, respectively. Some other reported compounds were found to be in the same range of free energy of binding, such as ZINC31166919 (−10.3 kcal/mol) and ZINC67912776 (−10.0 kcal/mol) [43]. A recent study showed that screening of natural compounds, such as phenyl glucuronide and hydrocortisone acetate, was able to inhibit HER2 with −6.63 and −5.41 kcal/mol, respectively [44]. The free energy of binding for the selected top 20 compounds are shown in Table 1. Taken together, the amino acid residues Leu726, Val734, Ala751, Lys753, Thr798, Gly804, Arg849, Leu852, Thr862, and Asp863 were found in common interaction as compared to the control compound Lapatinib.

Val734 and Leu852 were involved in hydrophobic interaction in both complex HER2-ZINC43069427 and HER2-ZINC95918662 as compared to HER2-Lapatinib. Leu726 was found in common interaction, which is consistent with the reported study. Lapatinib was reported to interact with HER2 through Leu726 [43] and this amino acid also was observed in the interaction of selected compounds. Selected ligands were found to interact nearly with the same amino acid residue as present in control compounds. Thus, our selected ligands (ZINC43069427 and ZINC95918662) followed the same pattern of having a binding affinity of −11.0 and −8.5 kcal/mol, respectively, which is more than the control ZINC1550477 (Lapatinib), having a binding affinity of −7.65 kcal/mol. Some interactions, like the H-bond and hydrophobic interactions, were found to play an important role in correcting the position of the ligands at the active pocket of the receptor.

Recent medication development studies have shown that naturally occurring polyphenolic chemicals can inhibit BC cell proliferation and induce apoptosis [45]. The compounds with the highest negative free energy of binding, as compared to the control, were forwarded for molecular dynamics simulation study (MDS). The selected complexes HER2-ZINC43069427 and HER2-ZINC95918662 were subjected to 50 ns MDS.

### 3.4. MDS Outcomes

The selected complexes HER2-ZINC43069427 and HER2-ZINC95918662 were forwarded for MDS due to the high binding affinity of selected ligands with the target. MDS was performed using the GROMACS module gmx_rmsd for the selected complex to check stability. The topology file of the ligands was obtained from PRODRG and used to run the MDS. The dynamic behavior of the complexes was analyzed by the RMSD and RMSF. The RMSD is a structural and dynamic measure that is used to examine the conformational stability of a complex. A higher RMSD value indicates that a protein complex is less stable, and vice versa [46]. In this study, the RMSD of the HER2-ZINC43069427 and HER2-ZINC95918662 complex concerning the backbone atom was plotted against the MDS time, as shown in Figure 3.

The average RMSD in the HER2-ZINC43069427 complex was 0.32 nm, with oscillations around 5 ns and 35 ns. The average RMSD of the HER2-ZINC95918662 complex is about 0.32 nm, and there is some variation before 10 ns, but it remains constant for the rest of the simulation duration. The lower average value and volatility of the RMSD in both complexes are practically identical, implying that the complex is stable and has a strong bonding. The backbone atoms of each amino acid residue of HER2 in the HER2-ZINC43069427 and HER2-ZINC95918662 complex are presented in Figure 4 as RMSF, which is a useful metric for estimating residue flexibility during dynamics.

In HER2-ZINC43069427, the average RMSF is about 0.12 nm. Likewise, in HER2-ZINC95918662, with an average RMSF of 0.12 nm, higher spikes can be seen at the same locations as shown in the plot of Figure 4.

The radius of gyration (Rg) denotes the degree of protein compaction. It’s the mass-weighted RMSD for a group of atoms measured from their shared center of mass. As a result, the Rg’s trajectory analysis displays the change of the protein’s overall dimension throughout dynamics. The average Rg value for HER2-ZINC43069427 was 1.88 nm, with a significant drop before 20 ns (Figure 5). In the instance of the HER2-ZINC95918662, on the other hand, the average value of Rg was about 1.92 nm. HER2-ZINC43069427 and HER2-ZINC95918662 had their solvent-accessible surface area (SASA) determined (Figure 6).

SASA may alter as a result of a drug’s interaction with a protein’s structural characteristics. A greater SASA score indicates that the protein structure is expanding. During the simulation, there should be little variation in the SASA value. HER2-ZINC43069427 showed a lower SASA value than HER2-ZINC95918662. The SASA value suggests ZINC95918662 might cause protein expansion, increasing the protein’s solvent-accessible surface, as illustrated in Table 4.

HER2 is linked to BC and is one of the most important targets for lowering the BC rate [15,47,48]. The compounds with higher dock scores than the reference compound were investigated for interactions with important amino acid residues, and MDS was performed. Furthermore, MDS has produced reliable RMSD, RMSF, Rg, and SASA. The number of hits was eventually reduced to two, based on connections between ligands and active site residues, molecular dock score, and stable MDS findings, identifying ZINC43069427 and ZINC95918662 as possible leads against HER2 BC management. For pharmaceutical research, computerized virtual screening provides a viable method for the discovery of new prospective medications [49]. Structure-based computing has been used to effectively find several new ligands [50,51,52]. Hence we explore the binding affinity of ZINC43069427 and ZINC95918662 with HER2 for the management of BC.

## 4. Conclusions

Before going to the experimental stage, in silico investigations can save a lot of time and money. In silico tools can help predict the probable active drug along with various ADMET aspects and toxicological effects. The current study employed several prediction techniques to forecast the oral bioavailability of substances, which might pave the way for the development of new, safer medications. With the help of the in-silico platform, we identified two natural compounds, ZINC43069427 and ZINC95918662, from the library of 80617 natural compounds obtained from the ZINC database. After evaluating the screening, molecular docking, and MDS studies, we found that ZINC43069427 and ZINC95918662 may be used as possible agents against HER2 for the management of BC. A further wet-lab study is required to further evaluate these selected compounds.

## Figures and Tables

**Figure 1 life-12-01729-f001:**
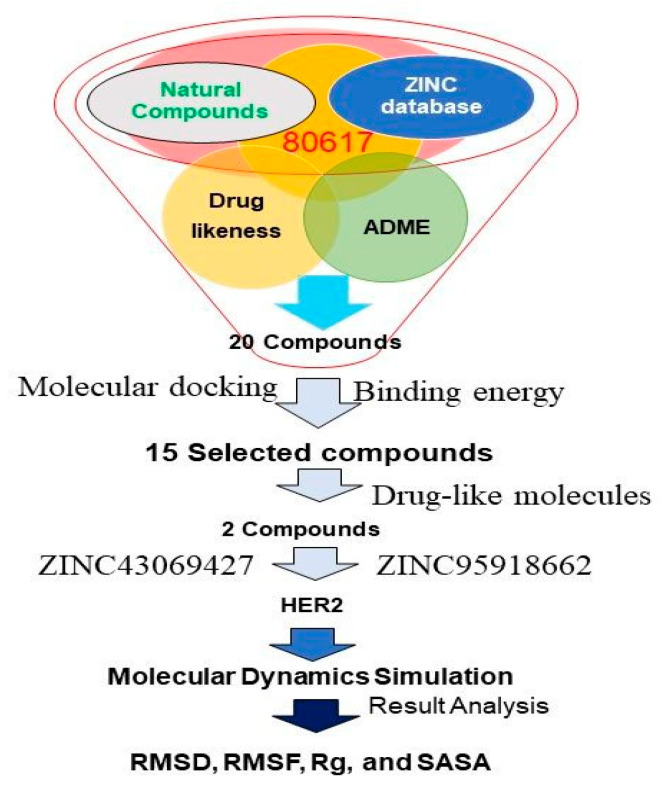
The different stages of the virtual screening process of natural compounds against HER2.

**Figure 2 life-12-01729-f002:**
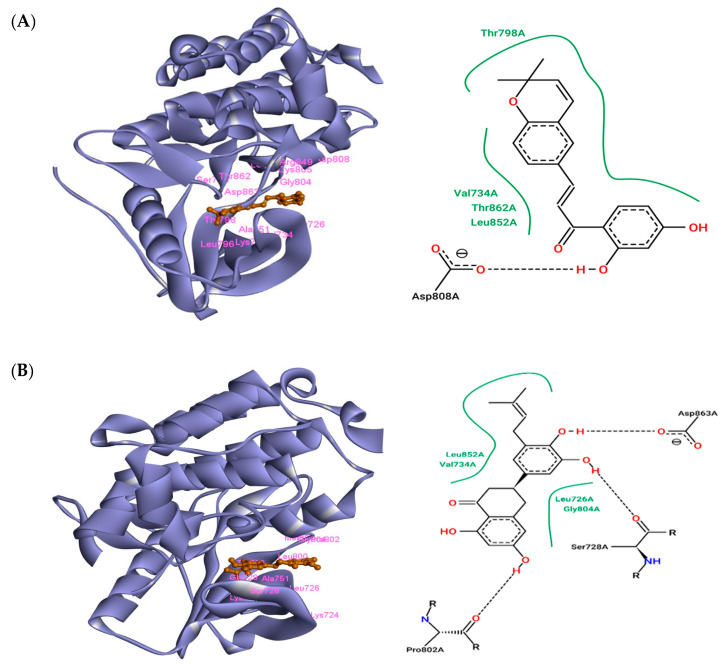
Interaction of HER2 with selected ligands and Lapatinib as a control. (**A**) 3D and 2D interaction of ZINC43069427 with HER2. (**B**) 3D and 2D interaction of ZINC95918662 with HER2. (**C**) 3D and 2D interaction of Lapatinib with HER2.

**Figure 3 life-12-01729-f003:**
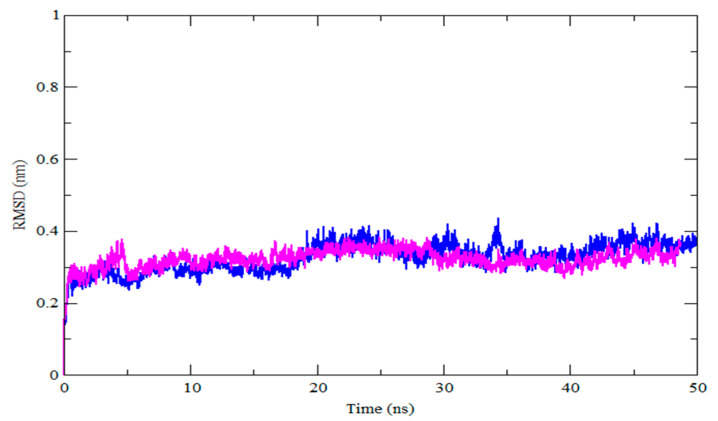
RMSD as a function of simulation time for HER2 with ZINC43069427 (blue color) and ZINC95918662 (magenta color).

**Figure 4 life-12-01729-f004:**
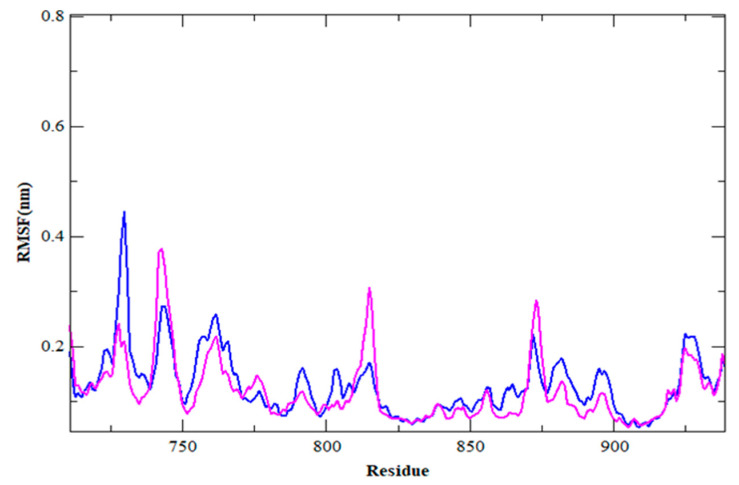
RMSF for HER2-ZINC43069427 (blue color) and HER2-ZINC95918662 complex (magenta color).

**Figure 5 life-12-01729-f005:**
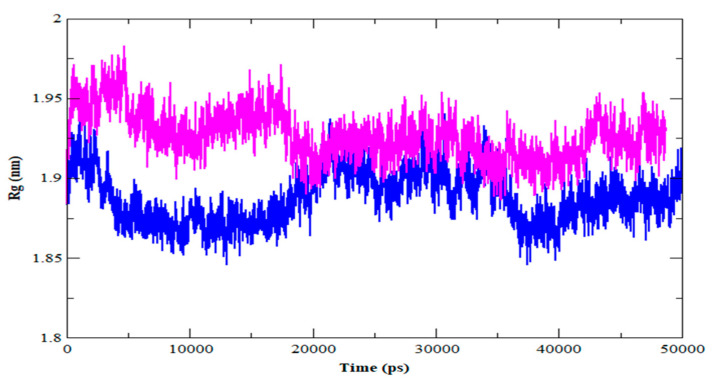
Radius of gyration (Rg) for HER2-ZINC43069427 (blue color) and HER2-ZINC95918662 complex (magenta color).

**Figure 6 life-12-01729-f006:**
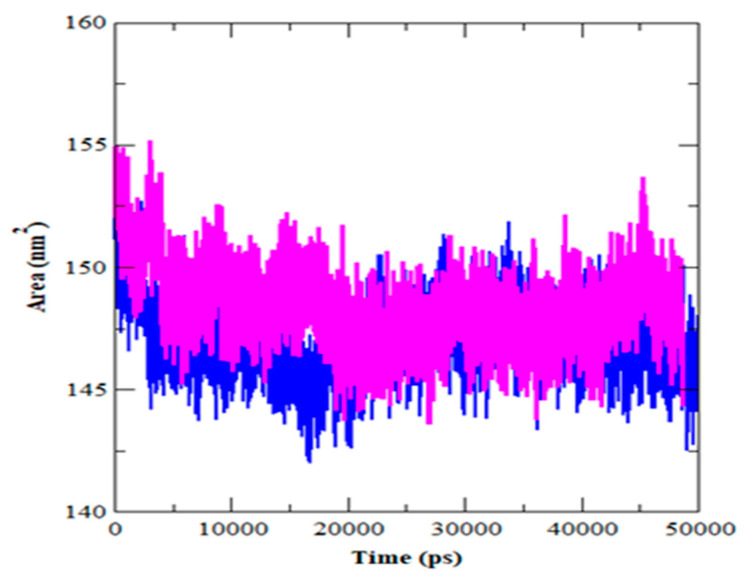
Solvent accessible surface area (SASA) as a function of simulation time for HER2-ZINC43069427 (blue color) and HER2-ZINC95918662 complex (magenta color).

**Table 1 life-12-01729-t001:** The binding energy of top selected compounds with HER2.

S.No.	List of Compounds	Binding Energy (kcal/mol)
1.	ZINC43069427	−11.0
2.	ZINC95918662	−8.5
3.	ZINC1550477 (Lapatinib)	−7.65
4.	ZINC3976838 (Afatinib)	−7.80
5.	ZINC34587071 (Sapitinib)	−7.15
6.	Salvianolic acid C	−7.10
7.	ZINC000000006256	−8.3
8.	ZINC000000000052	−8.1
9.	ZINC000000000056	−8.1
10.	ZINC000000000446	−8.0
11.	ZINC000000001288	−8.0
12.	ZINC000000001161	−7.9
13.	ZINC000095918662	−7.8
14.	ZINC000095918662	−7.5
15.	ZINC000095918662	−7.4
16.	ZINC000095918662	−7.2
17.	ZINC000095918662	−7.0
18.	ZINC000095918662	−6.8
19.	ZINC000000000290	−6.9
20.	ZINC000000006606	−6.7
21.	ZINC000000001288	−6.0
22.	ZINC000043069427	−5.8
23.	ZINC000043069427	−5.5

**Table 2 life-12-01729-t002:** ADMET properties for the selected ligands ZINC43069427 and ZINC95918662.

Property	Parameters					ZINC43069427	ZINC95918662
Absorption	Water solubility (log mol/L))					−4.135	−3.811
	Caco2 permeability					1.024	1.015
	Intestinal absorption (human) (% absorbed)					92.481	82.451
	Skin permeability (log Kp)					−3.121	−2.739
	P-glycoprotein	substrate			(Yes/No)	Yes	Yes
		I inhibitor				Yes	No
		II inhibitor				No	No
Distribution	VDss (human) (log L/kg)					0.064	0.036
	Fraction unbound (human) (Fu)					0.092	0.104
	BBB permeability (log BB)					−0.123	−0.991
	CNS permeability (log PS)					−1.954	−2.243
Metabolism	CYP	2D6	substrate		(Yes/No)	No	No
		3A4				No	No
		1A2	inhibitor			Yes	Yes
		2C19				Yes	Yes
		2C9				Yes	Yes
		2D6				No	No
		3A4				Yes	No
Excretion	Total clearance (log ml/min/kg)					0.143	0.202
	Renal OCT2 substrate (Yes/No)					No	No
Toxicity	AMES toxicity (Yes/No)					No	Yes
	Max. tolerated dose (human) (log mg/kg/day)					0.004	0.318
	hERG I inhibitor (Yes/No)					No	No
	hERG II inhibitor (Yes/No)					Yes	No
	Oral Rat Acute Toxicity (LD50) (mol/kg)					2.301	2.215
	Oral Rat Chronic Toxicity (LOAEL) (log mg/kg_bw/day)					1.793	2.16
	Hepatotoxicity				(Yes/No)	No	No
	Skin sensitisation					No	No
	*T. Pyriformis* toxicity (log ug/L)					0.83	0.38
	Minnow toxicity (log mM)					0.513	0.575
Physicochemical	Formula					C_20_H_18_O_4_	C_21_H_22_O_5_
	Molecular weight (g/mol)					322.35	354.40
	No. heavy atoms					24	26
	No. arom. heavy atoms					12	12
	No. rotatable bonds					3	3
	No. H-bond			acceptors		4	5
				donors		2	4
	Molar Refractivity					94.42	100.59
	TPSA (Å²)					66.76	97.99
Druglikeness	Lipinski					Yes	Yes
	Ghose					Yes	Yes
	Veber					Yes	Yes
	Egan					Yes	Yes
	Muegge					Yes	Yes

**Table 3 life-12-01729-t003:** Different rules for compounds of drug-like molecules.

Drug Like Properties	Cutoff Range	Nature of Selected Compounds	
		ZINC43069427	ZINC95918662
Lipinski Rule of Five	1. Molecular mass: <500 Dalton 2. High lipophilicity: LogP <5 3. Hydrogen bond donors: <5 4. Hydrogen bond acceptors: <10 5. Molar refractivity: 40 to130	Yes	Yes
Ghose Rule	1. clogP: −0.4 to 5.6 2. MW: 160 to 480 3. Molar refractivity: 40 to 130 4. Total number of atoms: 20 to 70	Yes	Yes
Veber Rule	1. Rotatable bonds: ≤10 2. Polar surface area: ≤140 Å^2^	Yes	Yes
Egan Rule	1. WlogP: ≤5.88 2. TPSA: ≤131.6	Yes	Yes
Muegge Rule	1. MW: 200 to 600 2. XlogP: −2 to 5 3. TPSA: ≤150 4. Number of rings: ≤7 5. Number of carbon: >4 6. Number of heteroatoms: >1 7. Number of rotatable bonds: ≤15 8. Hydrogen bond acceptors: ≤10 9. Hydrogen bond donors: ≤5	Yes	Yes

**Table 4 life-12-01729-t004:** The values of MDS parameters for the selected complexes HER2-ZINC43069427 and HER2-ZINC95918662.

Parameters	HER2-ZINC43069427	HER2-ZINC95918662
RMSD (nm)	0.32	0.32
RMSF (nm)	0.12	0.12
Rg (nm)	1.88	1.92
SASA	146.99	148.38

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
