# Peer review of "Screening, Docking, and Molecular Dynamics Study of Natural Compounds as an Anti-HER2 for the Management of Breast Cancer"

_life, 2022, doi:10.3390/life12111729_

Round 1
Reviewer 1 Report
-Improve the introduction by considering some some sort of focus on natural compounds.
-It is reported “Binding energy of top 20 comopounds.”, probably it is “comopounds”, in table 3
-Revise the line no 92, it is not clear
-Revise the line no 20 by adding future drug development
-Add RMSF in line no 26
-In Table 1, at the row “formula” put the number in subscript
-Include the ZINC ID of all compounds in table 3.
-Revise the legend for table 4.
-Improve the molecular dynamics simulation part of the result and discussion.
-Revise the sentence in line no. 66-68, and add the proper reference in line no82-83.
-Try to add aim of this study in Introduction of manuscript.
-Add some latest references to improve the manuscript.
-English improved required with spell check. Typos and grammar.
Author Response
Response: We (all authors) are grateful to the learned reviewer for giving his valuable time and providing critical comments to improve the quality of our manuscript. We have tried our best effort against each raised comment leading to changes/ corrections in the revised version of the manuscript. Hope these changes/corrections will meet your approval.
Comment 1- Improve the introduction by considering some sort of focus on natural compounds.
Response 1- It is added in the revised manuscript.
Comment 2- It is reported “Binding energy of top 20 comopounds.”, probably it is “compounds”, in table 3
Response 2- It is revised.
Comment 3- Revise the line no 92, it is not clear
Response 3- It is revised
Comment 4- Revise the line no 20 by adding future drug development
Response 4- It is revised
Comment 5- Add RMSF in line no 26
Response 5- It is added in the revised manuscript.
Comment 6- In Table 1, at the row “formula” put the number in subscript
Response 6- It is revised
Comment 7- Include the ZINC ID of all compounds in table 3.
Response 7- ZINC id of Sapitinib added and the ZINC id for Salvianolic acid C is not available.
Comment 8- Revise the legend for table 4.
Response 8- It is revised
Comment 9- Improve the molecular dynamics simulation part of the result and discussion.
Response 9- It is added in the revised manuscript.
Comment 10- Revise the sentence in line no. 66-68, and add the proper reference in line no82-83.
Response 10- It is revised and proper citations were added to make it more clear.
Comment 11- Try to add aim of this study in Introduction of manuscript.
Response 11- The aim of this study has been added in the revised manuscript.
Comment 12- Add some latest references to improve the manuscript.
Response 12- It is revised by adding recent references
Comment 13- English improved required with spell check. Typos and grammar.
Response 13- Manuscript has been revised to improve the errors and grammar.
Reviewer 2 Report
In this manuscript, the authors utilized docking to screen 80617 natural compounds from the ZINC library and identified two ligands that showed high binding affinity to the HER2. The ligands could potentially be used as inhibitors for targeting HER2 to cure breast cancer. The authors further performed molecular dynamics simulation to demonstrate the stability of the identified ligands in binding to HER2. Overall, it’s an interesting study and their computational results may help the identification of novel molecules for targeting HER2.
However, there are several major points that still need to be addressed. Specifically, the flow and the organization of the manuscript needs to be improved. Some of the figures need to be rearranged and additional structural figures need to be shown. Additional analysis and discussion is required to better interpret the results. A lot of English grammar errors in the current version of the manuscript need to be improved. I would recommend the authors to extensively revise their manuscript before considering the publication of this manuscript in Life.
Specific points:
1. The introduction requires significant improvement in the flow and the contents. I would suggest the authors elaborate on the introduction of HER2 and the role of HER2 in breast cancer. Also, the author should discuss previous work in chemically targeting HER2 for curing breast cancer as well as the remaining challenges in the field. The introduction of HER2 in the first paragraph of results and discussion can be moved to the introduction.
2. As the method details are at the end of the manuscript. It would be helpful if the authors could provide a method overview of how they identified the two selected ligands at the beginning of results and discussion. That will help improve the readability as it is unclear how the two ligands were selected when the readers get to the results section.
3. What are the error bars for those free energy of binding reported in the manuscript? This must be reported.
4. Figures 1-3 could be combined in a single figure. That will be helpful for the readers to compare the interactions between different ligands. In addition, the authors should provide 3D structural snapshots of the docked HER2-ligand complexes, which can be shown by the sides of current 2D interaction schemes. For the three 3D structural snapshots, HER2 should have the same orientation in order to compare the interactions between the three ligands. In that way, Figure 4 wouldn’t be needed or only the superposition of the compounds is necessary.
5. Are there any experimentally-solved structures available for the ligand-bound HER2 complex? It would be helpful to compare the docked ligand-bound complex structure with the experimental structures. That is to help validate the docking results.
6. Despite the interaction details provided in Table 4, what can we learn regarding the structural-functional relationship of the compounds for HER2? The authors should try to summarize the molecular origin of the differences in binding affinity for those ligands and the implications for rational design of compounds for targeting HER2. Do those two ligands have different scaffolds compared to the existing known inhibitors for HER2?
7. For RMSF values, it doesn’t make sense to provide numbers with 6 digits after the decima. Two digits would be sufficient. That applies to all the numbers in Table 5, too.
8. The authors should explain their steps for screening compounds. That should include all the parameters and protocols used in different stages of the virtual screening process. How are the initial 80617 compounds chosen? It would be helpful if the authors can draw a process diagram to clearly show how they screen the library and select the last two candidates.
9. “which was defined as a cubical box with a minimum distance of 10 from the protein surface.” Missing unit for the distance. I am assuming the unit is Angstrom.
10. There are a lot of very basic English grammar errors throughout the manuscript. The authors should carefully edit their manuscript to correct those errors. Some of the examples are below.
“The amino acid residues …… was found in common interaction” - “were” should be used. This error occurs in many places of the manuscript.
“HER2 has been shown to an important role in” - missing “play”
"was used in depth study" - "in further study"
Author Response
In this manuscript, the authors utilized docking to screen 80617 natural compounds from the ZINC library and identified two ligands that showed high binding affinity to the HER2. The ligands could potentially be used as inhibitors for targeting HER2 to cure breast cancer. The authors further performed molecular dynamics simulation to demonstrate the stability of the identified ligands in binding to HER2. Overall, it’s an interesting study and their computational results may help the identification of novel molecules for targeting HER2.
However, there are several major points that still need to be addressed. Specifically, the flow and the organization of the manuscript needs to be improved. Some of the figures need to be rearranged and additional structural figures need to be shown. Additional analysis and discussion is required to better interpret the results. A lot of English grammar errors in the current version of the manuscript need to be improved. I would recommend the authors to extensively revise their manuscript before considering the publication of this manuscript in Life.
Response: We (all authors) are grateful to the learned reviewer for giving his valuable time and providing critical comments to improve the quality of our manuscript. We have tried our best effort against each raised comment leading to changes/ corrections in the revised version of the manuscript. Hope these changes/corrections will meet your approval.
Specific points:
Comment 1: The introduction requires significant improvement in the flow and the contents. I would suggest the authors elaborate on the introduction of HER2 and the role of HER2 in breast cancer. Also, the author should discuss previous work in chemically targeting HER2 for curing breast cancer as well as the remaining challenges in the field. The introduction of HER2 in the first paragraph of results and discussion can be moved to the introduction.
Response 1: The introduction has been revised to make it more clear. The introduction of HER2 in the first paragraph of results and discussion have been moved to the introduction.
Comment 2: As the method details are at the end of the manuscript. It would be helpful if the authors could provide a method overview of how they identified the two selected ligands at the beginning of results and discussion. That will help improve the readability as it is unclear how the two ligands were selected when the readers get to the results section.
Response 2: Authors are thankful to the reviewer for their constructive feedback. It is revised accordingly to make more clear the manuscript.
Comment 3: What are the error bars for those free energy of binding reported in the manuscript? This must be reported.
Response 3: Authors are very thankful to the reviewers for their constructive feedback. We did docking analysis by Autodock tools and it just resulted the binding efficacy in terms of binding energy that we mentioned in the manuscript. We have the utmost respect for the erudite reviewer, but we are unable to comprehend the need for "error bars" given that many docking results in the literature lack them.
Comment 4: Figures 1-3 could be combined in a single figure. That will be helpful for the readers to compare the interactions between different ligands. In addition, the authors should provide 3D structural snapshots of the docked HER2-ligand complexes, which can be shown by the sides of current 2D interaction schemes. For the three 3D structural snapshots, HER2 should have the same orientation in order to compare the interactions between the three ligands. In that way, Figure 4 wouldn’t be needed or only the superposition of the compounds is necessary.
Response 4: Figures 1-3 has been combined in a single figure as per reviewer suggestion to make it more clear. Figure 4 have been removed.
Comment 5: Are there any experimentally-solved structures available for the ligand-bound HER2 complex? It would be helpful to compare the docked ligand-bound complex structure with the experimental structures. That is to help validate the docking results.
Response 5: Authors are thankful to the reviewer for their constructive feedback. We were trying to find experimentally-solved structures, but it is not available, so we used Crystal structure of HER2.
Comment 6: Despite the interaction details provided in Table 4, what can we learn regarding the structural-functional relationship of the compounds for HER2? The authors should try to summarize the molecular origin of the differences in binding affinity for those ligands and the implications for rational design of compounds for targeting HER2. Do those two ligands have different scaffolds compared to the existing known inhibitors for HER2?
Response 6: We used natural compounds for the inhibition of HER2. These two selected compounds are have different scaffolds, because these are natural compounds. Due to lack of natural therapy, we tried to identify new natural compounds as HER2 inhibitor.
Comment 7: For RMSF values, it doesn’t make sense to provide numbers with 6 digits after the decimal. Two digits would be sufficient. That applies to all the numbers in Table 5, too.
Response 7: These values have been revised throughout manuscript.
Comment 8: The authors should explain their steps for screening compounds. That should include all the parameters and protocols used in different stages of the virtual screening process. How are the initial 80617 compounds chosen? It would be helpful if the authors can draw a process diagram to clearly show how they screen the library and select the last two candidates.
Response 8: The screening process diagram has been added in the revised manuscript for better understanding. A complete library of natural compounds was obtained from ZINC database for screening process against HER2.
Comment 9: “which was defined as a cubical box with a minimum distance of 10 from the protein surface.” Missing unit for the distance. I am assuming the unit is Angstrom.
Response 9: It is mentioned in the revised manuscript.
Comment 10: There are a lot of very basic English grammar errors throughout the manuscript. The authors should carefully edit their manuscript to correct those errors. Some of the examples are below.
“The amino acid residues …… was found in common interaction” - “were” should be used. This error occurs in many places of the manuscript.
“HER2 has been shown to an important role in” - missing “play”
"was used in depth study" - "in further study"
Response 10: Manuscript has been revised to improve the grammar errors
Round 2
Reviewer 2 Report
While the authors properly addressed some of the reviewer's comments, they ignored a few major points raised by the reviewer. I would request the authors to revise their manuscript again to address the remaining points before considering the publication of this manuscript in Life.
(1) The authors should provide very clear 3D structural snapshots of the docked HER2-ligand complexes, which can be shown along with the current 2D interaction schemes. In the revised manuscript, Figure 2A is not clear for comparing the interactions. It's better to make separate figure for each HER2-ligand complex. The structural snapshots should highlight the HER2 binding pocket and the HER2-ligand interactions. HER2 should have the same orientation in order to compare the interactions between the three ligands.
(2) Despite the interaction details provided in Table 4, what can we learn regarding the structural-functional relationship of the compounds for HER2? The authors should summarize the molecular origin of the differences in binding affinity for those ligands and the implications for rational design of compounds for targeting HER2. Please add discussion on those points in your results and discussion.
(3) The error bars of the binding free energies could be computed from the standard deviation of multiple docking runs for the same ligand. The error bar is necessary for binding energy prediction.
Author Response
While the authors properly addressed some of the reviewer's comments, they ignored a few major points raised by the reviewer. I would request the authors to revise their manuscript again to address the remaining points before considering the publication of this manuscript in Life.
Response: We (all authors) are grateful to the learned reviewer for giving his valuable time and providing critical comments to improve the quality of our manuscript. We have tried our best effort against each raised comment leading to changes/ corrections in the revised version of the manuscript. Hope these changes/corrections will meet your approval.
Comment (1): The authors should provide very clear 3D structural snapshots of the docked HER2-ligand complexes, which can be shown along with the current 2D interaction schemes. In the revised manuscript, Figure 2A is not clear for comparing the interactions. It's better to make a separate figure for each HER2-ligand complex. The structural snapshots should highlight the HER2 binding pocket and the HER2-ligand interactions. HER2 should have the same orientation in order to compare the interactions between the three ligands.
Response: All the figures under Figure 2 have been revised accordingly. Figures are separately represents in the revised manuscript. Binding amino acid residues are mentioned in the 3D complex structure. We tried to put the structure in the same orientation. The exact orientation is impossible to put because different ligands interact with a few other amino acid residues (surrounding to binding pocket) so it makes changes the orientation.
Comment (2): Despite the interaction details provided in Table 4, what can we learn regarding the structural-functional relationship of the compounds for HER2? The authors should summarize the molecular origin of the differences in binding affinity for those ligands and the implications for rational design of compounds for targeting HER2. Please add discussion on those points in your results and discussion.
Response: Table 4 has been removed and the detailed amino acid residues were mentioned in the result and discussion part of the revised manuscript for better understanding. Selected ligands were found to interact nearly with the same amino acid residue as present in control compounds. So, our selected ligands (ZINC43069427 and ZINC95918662) followed the same pattern having binding affinity of -11.0 and -8.5 kcal/mol respectively, which is more than the control ZINC1550477 (Lapatinib) having binding affinity of -7.65 kcal/mol. Some interaction like H-bond and hydrophobic interaction was found to play an important role to correct the position of the ligands at the active pocket of the receptor.
Comment (3): The error bars of the binding free energies could be computed from the standard deviation of multiple docking runs for the same ligand. The error bar is necessary for binding energy prediction.
Response: We do respect to the esteemed reviewer for their feedback.
Until now, all the published papers have mentioned only the highest negative free energy of binding for the best complex, so we consider only these values.
The actual binding energy considered at zero RMSD was -11.0 and -8.5, and -7.65 kcal/mol for ZINC43069427 and ZINC95918662, and Lapatinib, respectively. for the ten runs obtained from docking, the average binding energy for these compounds was found as -7.71, -9.81, and -7.71 kcal/mol. The standard deviation was calculated and found as ±0.14, ±0.70, and ±1.41.
The error bars of the binding free energies were computed from the standard deviation of multiple docking runs for the ligands ZINC43069427 and ZINC95918662, and Lapatinib. It was found as -11.0±0.14 and -8.5±0.70, and -7.65±1.41 kcal/mol for ZINC43069427 and ZINC95918662, and Lapatinib, respectively.